# Immunological Aspects Related to Viral Infections in Severe Asthma and the Role of Omalizumab

**DOI:** 10.3390/biomedicines9040348

**Published:** 2021-03-30

**Authors:** Francesco Menzella, Giulia Ghidoni, Carla Galeone, Silvia Capobelli, Chiara Scelfo, Nicola Cosimo Facciolongo

**Affiliations:** Pneumology Unit, Azienda Unità Sanitaria Locale-IRCCS di Reggio Emilia, 42122 Reggio Emilia, Italy; giulia.ghidoni@ausl.re.it (G.G.); carla.galeone@ausl.re.it (C.G.); silvia.capobelli@ausl.re.it (S.C.); chiara.scelfo@ausl.re.it (C.S.); nicolacosimo.facciolongo@ausl.re.it (N.C.F.)

**Keywords:** viral respiratory infections, severe asthma, immune response, biologicals, omalizumab

## Abstract

Viral respiratory infections are recognized risk factors for the loss of control of allergic asthma and the induction of exacerbations, both in adults and children. Severe asthma is more susceptible to virus-induced asthma exacerbations, especially in the presence of high IgE levels. In the course of immune responses to viruses, an initial activation of innate immunity typically occurs and the production of type I and III interferons is essential in the control of viral spread. However, the Th2 inflammatory environment still appears to be protective against viral infections in general and in severe acute respiratory syndrome coronavirus 2 (SARS-CoV-2) infections as well. As for now, literature data, although extremely limited and preliminary, show that severe asthma patients treated with biologics don’t have an increased risk of SARS-CoV-2 infection or progression to severe forms compared to the non-asthmatic population. Omalizumab, an anti-IgE monoclonal antibody, exerts a profound cellular effect, which can stabilize the effector cells, and is becoming much more efficient from the point of view of innate immunity in contrasting respiratory viral infections. In addition to the antiviral effect, clinical efficacy and safety of this biological allow a great improvement in the management of asthma.

## 1. Introduction

Viral respiratory infections are recognized risk factors for the loss of control of allergic asthma and the induction of exacerbations, both in adults and children [1,2,3]. Almost 80% of asthma exacerbations were associated with viral infections [1,2,3]. Severe asthma is more susceptible to virus-induced asthma exacerbations, especially in the presence of high IgE levels [4,5]. Respiratory viruses mainly involved in asthma exacerbations are human rhinoviruses (HRV), respiratory syncytial viruses (RSV), influenza viruses, parainfluenza virus, metapneumovirus, bocavirus, adenovirus, coronavirus [1,3,6]. HRV are the most common causative agents found in the 5 days preceding the development of exacerbations [3]. Several studies showed seasonal variations in hospital admissions for asthma related to viral upper respiratory tract infections, with higher rates of asthma exacerbations in autumn (28.8%) and spring (19.6%) [3]. The frequency of asthma exacerbations in autumn, winter and spring was 1.5–2 times higher than the one observed in summer in both adults and adolescents, if not adequately treated [7]. The annual peak of paediatric hospitalizations for asthma was on September, 7–10 days after the reopening of schools, and an additional peak, albeit less pronounced, was during the spring season [3].

These observations reinforce the idea that asthma exacerbations can be promoted by viral respiratory infections [3]. Asthmatic exacerbations consist of episodes characterized by an increase in respiratory symptoms (dyspnoea, cough, wheezing, chest tightness) and worsening of lung function requiring changes of therapy, the intake of oral corticosteroids (OCS), emergency treatment, and hospitalization in the most severe cases [2,3]. In terms of pulmonary function tests, asthma exacerbation is defined as a >20% reduction in FEV_1_ (Forced Expiratory Volume in 1 s) from baseline or a >30% reduction in peak expiratory flow (PEF) for 2 consecutive days. Asthmatic exacerbations can accelerate the decline in lung function and increase the risk of subsequent exacerbations [2,3,8].

In addition to the extra costs of managing the disease, asthma exacerbations can impair school attendance or patients’ ability to work and lead to increased asthma mortality [1,2,3]. According to the World Health Organization’s Report on Chronic Respiratory Diseases, approximately 250,000 deaths from asthma occur each year which, in many cases [9], could be avoided by better therapeutic management of the underlying disease and exacerbations [3,10].

Viral infections can cause asthma exacerbations through multiple mechanisms [3,5,6]: increased serum IgE levels, promotion of respiratory reactivity (including through epithelial damage), increased eosinophilic inflammation of the respiratory tract, increased neutrophilic inflammation of the lower airways, direct infection of the lower respiratory tract.

In this review, we analyze mechanisms underlying the immune response to respiratory viruses in patients with asthma and the effect of omalizumab in modulating the immune system by reducing virus-induced asthma exacerbations, including severe acute respiratory syndrome coronavirus 2 (SARS-CoV-2).

## 2. Methods

We conducted an extensive literature search using validated keyword filters to select articles related to viral respiratory infections in severe asthma and related immune response. The research was conducted on articles published from 1 January 2000 to 28 February 2021, and an in-depth and selective search was performed on biomedical bibliographic databases (MEDLINE (PubMed): https://pubmed.ncbi.nlm.nih.gov/, EMBASE: https://www.embase.com/#search/, SCOPUS: https://www.scopus.com/search/form.uri?display=basic#basic accessed on 28 February 2021) and Google Scholar: https://scholar.google.com/. Research papers, international guidelines and meta-analyses were considered, as well as articles published “ahead of print”. Because of the lack of randomized clinical trials (RCTs), reviews, case reports and case series were also included. The following keywords were used: refractory asthma, eosinophilia, allergy, therapies, inflammation, immune response, biologics, omalizumab, COVID-19, cytokines, and interleukins. Finally, the best and most authoritative documents were also considered based on chronological parameters, in particular the date of publication, in order to set up a defined research path and a coherent development of the arguments supporting the conclusions.

## 3. Immune Response to Respiratory Viruses and SARS-CoV-2

During immune responses to viruses, an initial activation of innate immunity typically occurs and the production of type I and III interferons (IFN-α/β and -λ, respectively) is essential in the control of viral spread. Plasmacytoid dendritic cells (pDCs) are the primary blood source of IFN-α [11]. It is known that a reduction in the production of IFN by pDCs and epithelial cells is frequent in atopic patients, with a consequent delayed and inefficient antiviral defence [12]. Another important aspect is that the intensity of the response to IFN-α after the ex vivo viral challenge of pDCs is inversely related to serum IgE levels. This evidence suggests that pDCs antiviral responses may be suppressed in atopic patients. In fact, asthmatics show great susceptibility to respiratory viral infections, which are often a trigger for exacerbations [13]. In a fundamental and refined study about paediatric patients, targeted short-term treatments with omalizumab, an anti-IgE recombinant humanized monoclonal antibody for the treatment of severe allergic asthma, improved IFN-α responses to HRV. There, within the omalizumab treatment group, major increases in IFN-α were associated with a significant reduction in the number of exacerbations [14]. These findings reiterated the role of compromising innate immunity in susceptibility to viral infections as a cause of asthma exacerbations and highlighted the fact that some biologics can solve or mitigate this problem.

However, the inflammatory Th2 environment appears to be able to favor viral infections, through the altered release of IFN-α in the peripheral blood mononuclear cells (PBMCs) of asthmatic patients, the reduction of NF-kB and the lack of activation of Toll-Like Receptor (TLR)7 and TLR8. In contrast, Th2 inflammation appears protective against SARS-CoV-2 infection [15]. It has been hypothesized that a greater number of airway eosinophils in asthmatic patients may be protective against the exaggerated inflammatory responses of the severe Coronavirus Disease 19 (COVID-19) phenotype, possibly because it reduces the hyperinflammation present in the advanced stage that generally marks severe respiratory viral diseases [16]. It is important to remember that distinctive immunological alterations have been observed during COVID-19: in particular in the plasma of severe patients, inflammatory cytokines (IL-6, TNF-α, IL-10, IL-2, IL-7, CXCL10, CCL2, CCL3) are present in much higher concentrations (“cytokine storm”). These aspects are associated with inflammation, parenchymal damage, and pulmonary infiltrates with extensive lung tissue damage [17,18].

Furthermore, the real role of eosinophils in the context of infections has not been precisely established yet. Eosinophils are reduced in peripheral blood from SARS-CoV-2 infected patients [19]. Eosinophilic cells blood count seems to have a major role in COVID-19 diagnosis and prognosis. Eosinopenia has been associated in up to 81% of cases and was proposed as a possible diagnostic marker for the disease. Persistent eosinopenia was associated with higher mortality [20]. A realistic hypothesis is that the increase in the number of eosinophils in the airways of asthmatic patients could be protective against excessive inflammatory responses in cases of severe COVID-19. In preclinical studies, it was shown that eosinophils are equipped with a series of molecular tools that allow them to recognize, respond and orchestrate antiviral responses in the airways [21]. There are very recent data about eosinophils regarding COVID-19, which although limited does not indicate that patients with eosinophil-associated diseases will have an altered course of COVID-19 [22]. This occurs in the case of subjects not immunosuppressed by concomitant medications such as high dose corticosteroids or immunosuppressants or their primary disease process. Similarly, although some preclinical studies have provided convincing experimental evidence that eosinophils have potential antiviral activity, there is no evidence that patients with eosinopenia induced by recently approved anti-eosinophil therapies have an increased susceptibility to viruses. There are reports of patients treated with anti-eosinophil drugs such as benralizumab with concomitant COVID-19 who have had a very mild disease course [23]. It has not yet been proven with certainty whether COVID-19-associated acquired eosinopenia contributes directly to the course of the disease, but it should be noted that pulmonary eosinophilia is not part of the secondary lung disease due to SARS-CoV-2 infection [22].

Numerous and recent studies showed that eosinopenia was present in patients with severe COVID-19 phenotypes and in many cases with fatal evolution [21,22,23,24].

The pathophysiology of eosinopenia in COVID-19 has not been fully clarified yet, but it is probably multifactorial. Possible explanations concern the inhibition of eosinophils egress from the bone marrow, the blockage of eosinophilopoiesis, the reduced expression of chemokine receptors and adhesion factors and finally the direct eosinophilic apoptosis induced by type 1 IFN released during acute infection [25,26].

## 4. Severe Asthma and COVID-19: Epidemiological Data

As for now, literature data, although limited and preliminary, show that severe asthma patients treated with biologics don’t have an increased risk of SARS-CoV-2 infection or progression to severe forms compared to the non-asthmatic population [27]. A survey conducted in the period 1–20 April 2020, in which six Italian clinical centers participated for a total of 473 adult patients receiving biologics (mean age 52 ± 12 years; 30.6% omalizumab), indicated that the prevalence of SARS-CoV-2 infection among patients with severe asthma was very low and it was comparable to the general population (0.8%; 95% CI 0.230–2.150) [27]. Briefly, in this study four patients with a confirmed diagnosis of COVID-19, three on omalizumab therapy for 60, 72, and 96 months (critical/severe/mild) and one on benralizumab for 6 months (mild), went into remission. A second Italian survey examined the use of anti-IgE and anti-IL5 biological therapies in children/adolescents with severe allergic asthma and other forms of atopy during the pandemic period (20 participating centers, 308 asthmatic patients on biologic therapy, monitored between February and April 2020). It showed that only three patients (1%) developed COVID-19. All COVID-19 cases were paucisymptomatic and resolved quickly, with no hospitalization or worsening of the underlying disease [28]. These data, in agreement with other international assessments, indicate that treatments with biologics during the COVID-19 pandemic appear to be safe and can be continued in patients with severe allergic asthma, interrupting it only in case of confirmed SARS-CoV-2 positivity [28] and up to negativity of the nasopharyngeal swab [27,28,29]. Furthermore, the latest published SANI (Severe Asthma Network in Italy) registry data indicate that, despite the regular use of OCS by the majority of adolescent and adult patients with severe asthma, in many cases a high degree of systemic inflammation remains indicative of a probable partial response to OCS [29] (Table 1). Therefore, the Global Initiative for Asthma (GINA) guidelines state that a short course with OCS is useful in the exacerbation phase of severe asthma, but it must subsequently be reduced/stopped due to the serious side effects associated with it. Moreover, they affirm that it is important to consider the use of biologicals [30,31]. Especially at this time of the global pandemic, the use of biologics for severe refractory asthma could reduce the use of OCS and their possible negative clinical impact [32,33].

### 4.1. Impact of Monoclonal Antibodies for Severe Asthma on Viral Infections

Considering these observations, it is natural to ask whether therapies with biologics, such as anti IL-5 or anti-eosinophilic monoclonal antibodies (mAbs), are potentially risky for viruses, in particular SARS-CoV-2. To clarify these doubts, we do not have many certainties at the moment, except for a few studies. The MATERIAL study had the main purpose to determine if mepolizumab 750 mg intravenous (IV) could change the immune and inflammatory responses to rhinovirus type 16 (RV16) in mild asthma. The results showed that mepolizumab had no clinical effects in terms of immune response to this infection [34]. As for now, no studies have been published on patients with severe asthma specifically treated with mAbs except for a case study. In this article, a patient with severe eosinophilic asthma, who had been receiving benralizumab for two years, had a confirmed SARS-CoV-2 infection. The course of COVID-19 was very mild, with a rapid and complete resolution of the clinical picture [35].

Because of the increased susceptibility to respiratory viruses and virus-induced exacerbations and for the lack of data on patients with severe asthma treated with biologics in the first months of the COVID-19 pandemic, it was feared that patients with asthma could more easily experience SARS-CoV-2 infections with possible severe evolution and loss of underlying disease control [26,27]. Although still limited, epidemiological and real-life evidence, collected at national and international levels on various series of adult and paediatric patients, including those with severe allergic asthma receiving biologic therapies, provided reassuring indications [26,27,28,29,30]. To date, there is no evidence of impaired immune response to SARS-CoV-2 in patients with severe asthma on biologic therapies. In placebo-controlled clinical trials, none of the monoclonal antibodies in use (including omalizumab) was associated with increased susceptibility of treated patients in experiencing viral infections or immunosuppression [26,30]. Conversely, the discontinuation of biologic therapy in patients with severe asthma, well controlled by treatment, may increase the risk of asthma exacerbations, resulting in a greater need for OCS, access to emergency room services and hospital admissions: all factors that increase the risk of exposure or infection to SARS-CoV-2 [30]. In the light of the available evidence, the main national and international scientific societies and the GINA 2020 guidelines agreed on the need not to interrupt ongoing treatments for the control of asthma during the COVID-19 pandemic [31,32,33].

This recommendation covers both inhaled drugs (including ICS) and biologics used as add-on therapies for the treatment of severe asthma, helping to reduce the risk of exacerbations and the use of OCS [4,32,33].

In particular, during the COVID-19 pandemic, treatment with biologics should be:Continued in uninfected patients with severe asthma who are benefiting from it [32,33].Maintained or temporarily suspended (only for what is strictly necessary) in case of SARS-CoV-2 infection, until complete clinical and virological remission [32,33].Preferably administered at home (home-use), with the aim to reduce access to health facilities/contacts at risk and to guarantee regular therapy intake [31,33,36,37].

### 4.2. Effect of Omalizumab in Patients with Severe Allergic Asthma and Respiratory Viral Infections

Omalizumab was the first biologic drug authorized for clinical use for the treatment of severe allergic asthma refractory to maximally dosed standard therapies. This recombinant humanized monoclonal antibody binds to the two Cϵ3 domains of the constant portion of IgE, thus forming IgE/anti-IgE immune complexes that prevent the interaction of IgE with high affinity (FcϵRI) and low affinity (FcϵRII/CD23) membrane receptors [37,38]. As a major effect of this mechanism, omalizumab inhibits all IgE-dependent cellular and molecular events involved in the immunological pathway of allergic asthma [39]. Numerous systematic reviews and pooled analyses of randomized controlled trials showed that omalizumab was able to significantly reduce the rate of asthma exacerbations even in the long term (up to 60 weeks of treatment) [40,41]. These results have also been confirmed and amplified by several real-life studies [42], which demonstrated significant improvements in symptom control, quality of life and reduction in the need and dose of (OCS) [42,43,44].

In addition to the effects described above, recent studies showed that omalizumab was able to improve the antiviral response in subjects with allergic asthma, otherwise inadequate; these patients are more susceptible to virus-induced asthma exacerbations, especially those with elevated IgE levels [28]. In fact, omalizumab showed to reduce virus-induced seasonal exacerbations in both adults and children [7]. The mechanism of action underlying the clinical efficacy of omalizumab involves the binding of omalizumab to IgE induced by allergen exposure, preventing its interaction with FcεRI and FcɛRII/CD23 located on the surface of different types of immune/inflammatory cells and structural airway cells [45,46].

In this way, omalizumab reduces the amount of free IgE that can trigger the allergic cascade, resulting in a decrease in the risk of severe exacerbations, asthma-related emergency visits, the need for OCS courses and a significant improvement in the quality of life (QoL), symptoms and lung function [47,48].

By sequestering IgE, omalizumab also positively influences aspects of the innate immune response to viral respiratory infections, which is deficient in patients with severe allergic asthma [49,50,51]. People with severe allergic asthma are more susceptible to respiratory viruses (85% of children and 50% of adults) and more likely to experience virus-induced asthma exacerbations, which are more severe in the presence of elevated IgE levels [48,49,50,51,52,53]. The action of Omalizumab on the antiviral response in patients with severe allergic asthma primarily involves pDCs. These cells play a key role in targeting the immune response through the rapid and massive production, stimulated by TLRs, of type I IFN (in particular, IFN-α) following infections [47,48,49,50] (Figure 1).

Virus-induced IFN-α responses are inversely related to FcεRI expression and further inhibited by receptor cross-linking. These data suggest a counter-regulation of the FcεRI and IFN-α pathways on pDCs and a reduction in the antiviral response due to the cross-linking of IgE induced by allergens [50,51,52,53]. Compared to healthy subjects, patients with allergic asthma show a greater expression of FcεRI on pDCs, associated with IgE levels, which correlates with a marked reduction in TLR expression and a lower amount of IFN-α in response to viral infections [47]. In atopic subjects, the treatment with omalizumab, in addition to directly counteracting the allergic response by sequestering IgE, indirectly reduces the number of FcεRI receptors located on basophils [45,47], mast cells and pDC [40,41,42]. Therefore, the net effect of omalizumab in patients with severe allergic asthma consists in favouring a better antiviral response mediated by TLR receptors and the increased production of IFN-α by pDC [48,50]. Omalizumab’s promotion of antiviral response was confirmed by studies evaluating the immune response to HRVs (major contributors to virus-induced asthma exacerbations) and influenza viruses in patients with severe allergic asthma who were receiving omalizumab or placebo [28,43]. In both cases, the omalizumab treatment was associated with increased production of IFN-α by pDC and a reduction in surface-exposed FcεRI receptors, which was significantly associated with a lower frequency of asthma exacerbations [28]. This evidence helps to explain why the treatment with omalizumab in children with severe allergic asthma, initiated 4–6 weeks before school return and maintained for 4 months, can contain the fall peak in HRV asthma exacerbations [54]. Asthma exacerbations are induced for more than 80% by viral infections, mainly attributable to HRV (two thirds of cases) and by influenza virus, on which omalizumab was also effective [28].

Omalizumab causes a reduction in the receptors present on dendritic cells (FcεRI), by blocking free IgE, whose role is fundamental in allergic asthma, thus restoring the ability to produce IFN-α [14,28,54,55]. This mechanism of action translates into a greater antiviral response, which contributes to several benefits: reduction in asthma exacerbations (including severe ones) and emergency visits related to asthma, less need to take OCS, significant improvement in asthma-related QoL [28,37] (Figure 1).

### 4.3. Omalizumab and Reduction of Virus-Induced Seasonal Exacerbations: Trial Data

Patients with severe asthma and viral infections (not only in paediatric age) appear more susceptible to exacerbations, especially during fall, leading to significant consequences: greater morbidity, higher healthcare costs and disease progression [14]. While implementation and adherence to asthma guidelines improve disease control, some patients continue to experience exacerbations [14]. Therefore, innovative and targeted therapeutic approaches are needed to reduce the frequency of these events [14]. The efficacy of omalizumab in improving antiviral response and consequently reducing potentially virus-induced seasonal exacerbations has been demonstrated in several clinical trials conducted in both adult/adolescent patients and paediatric populations with moderate-to-severe allergic asthma.

#### 4.3.1. Trial on Paediatric Patients: Focus on the PROSE Study

The PROSE study (Preventative Omalizumab or Step-up Therapy for Fall Exacerbations) was designed with the aim of evaluating the efficacy of omalizumab in a paediatric population, aged 6–17 years, when administered 4–6 weeks before returning to school and up to 90 days after the beginning [14]. The results showed that, in patients in treatment as step 5 severe allergic asthma (fluticasone propionate/salmeterol 500/50 µg bid), the addition of omalizumab reduced the percentage of patients with at least one exacerbation in the autumn period by 63%, compared to the addition of placebo (Table 2).

Of the 75 reported exacerbations that presented ≥1 nasal sample, 89% were associated with viral infections, of which 81% induced by HRV. In patients with more severe allergic asthma (step 5), omalizumab reduced virus-induced seasonal asthma exacerbations by 65% (OR 0.35; 95% CI 0.15–0.85) compared to placebo [14].

The efficacy of omalizumab was confirmed in a further prospective, observational cohort study on 161 patients aged 6–17 years with allergic asthma and for whom HRV-positive nasal samples were collected upon admission to emergency departments. In these children and adolescents, the treatment with omalizumab resulted in a greater reduction in the severity of rhinovirus-induced exacerbations than in those treated with ICS, despite patients in the omalizumab group had greater disease activity at baseline [6]. In particular, omalizumab treatment was associated with:62% reduction in the overall time of taking salbutamol every 2 h (15 h vs. 30.8 h; *p* < 0.001).42% reduction in hospital stay (34.5 h vs. 58.5 h; *p* < 0.001).

The efficacy of seasonal omalizumab treatment was confirmed in a Cochrane review [41], which concluded that:It is the only pharmacological strategy for asthma or intervention with evidence of efficacy in reducing autumn exacerbations in children with allergic asthma [55].It is particularly effective in patients with severe allergic asthma, for whom there are limited therapeutic options and who are more exposed to the risk of exacerbations [55].There is no evidence that omalizumab is associated with adverse events greater than placebo, except for injection site reactions [56].

#### 4.3.2. Trial on Adult and Adolescent Patients

Even in adult patients with severe allergic asthma not controlled by maximally dosed ICS/LABA therapy, omalizumab treatment showed to improve the innate antiviral response to influenza A and HRV by increasing the production of IFN-α. and IFN-λ [56].

The increased production of IFN-α and IFN-λ in 9 out of 10 patients treated with omalizumab for 6 months was statistically significant against influenza virus A, it showed a positive trend versus HRV and it was accompanied by an improvement in control of asthma symptoms [57].

In a post-hoc analysis of two phase III clinical trials, treatment with omalizumab showed to reduce the rates of autumn, winter and spring exacerbations in adolescent and adult patients with moderate-to-severe allergic asthma compared to placebo [7]. In fact, patients treated with placebo had a rate of flare-ups in autumn (7.2%) and winter (8.4%) twice higher, in spring (4.6%) about 1.5 times higher than the one observed during summer (2.9%). This trend was not observed for omalizumab (3.4% in autumn, 3.0% in winter, 1.5% in spring and 3.6% in winter). The probability of seasonal exacerbations with omalizumab was indeed 55% lower than with placebo (*p* = 0.0002) [7].

## 5. Conclusions

All mAbs currently used for the treatment of severe asthma are considered to be safe even in this historical moment with an important risk of viral infections, in particular from the SARS-CoV-2. Despite the fears that emerged during the initial stages of the pandemic, asthma is not a disease that carries a risk of developing severe forms of COVID-19. The growing evidence available in the literature is reassuring in this regard. Biologics for severe asthma are all effective, they are carefully chosen and are safe for viral infections. An important and still not fully understood problem is the management of patients with severe asthma during the COVID-19 pandemic. Clinicians should follow the recommendations of current evidence-based guidelines to prevent loss of control and exacerbations, unless further data emerge from the literature that could alter our understanding of the relative safety of drugs indicated in patients with asthma during this pandemic. Furthermore, as the absence of data indicating potential harms, current indications are not to discontinue biological therapies during the COVID-19 pandemic in patients with asthma for whom such therapies are clearly indicated and have been effective. For patients with severe asthma infected with SARS-CoV-2, the decision to maintain or postpone biological therapy until the patient is cured should be a case-by-case decision supported by a multidisciplinary team. It would also be useful to establish a registry of cases of patients with severe asthma who have developed COVID-19, including those treated with biologics, in order to clarify the questions that are still open in this area. Unlike all other biologics for asthma currently available, for which there are currently no data on the possible effect of modulation of the antiviral response, omalizumab is the only biologic with proven antiviral action against viruses responsible for upper respiratory tract infections and asthma exacerbations. This is crucial at the time of the COVID-19 pandemic, not only because a better asthma control reduces access to health facilities. The challenges in the management of asthma patients that emerged during the pandemic highlight the need for a multidisciplinary approach in the indication and management of biologics in asthma even more evident, mainly involving allergists, pulmonologists, infectious disease specialists and paediatricians. The antiviral effect of omalizumab, now proven by significant evidence from the literature, is an extremely important aspect of this drug and it has not been found in any other biological drug for asthma at the moment. Omalizumab exerts a profound cellular effect, as it is able to stabilize the effector cells which become much more efficient from an innate immunity perspective in contrasting respiratory viral infections. It has an evident clinical impact in terms of improving asthma control and reducing the use of drugs, burdened by important and numerous side effects such as OCS. The quality of life consequently improves, not only for the reduction of exacerbations but also for the prevention or minimization of organ damage induced by OCS, which also have a high impact on health and in economic terms. In addition to the antiviral effect, the clinical efficacy and safety of this biological allow a great improvement in the management of asthma and lead to a great increase in the quality of life and productivity of patients. The big open problem of omalizumab, like other biologics, is the high direct cost, so it would be desirable to reduce it to guarantee the access of this effective and safe drug to a greater number of patients, also considering the ability to improve the immune response to viruses in frail asthmatic patients.

## Figures and Tables

**Figure 1 biomedicines-09-00348-f001:**
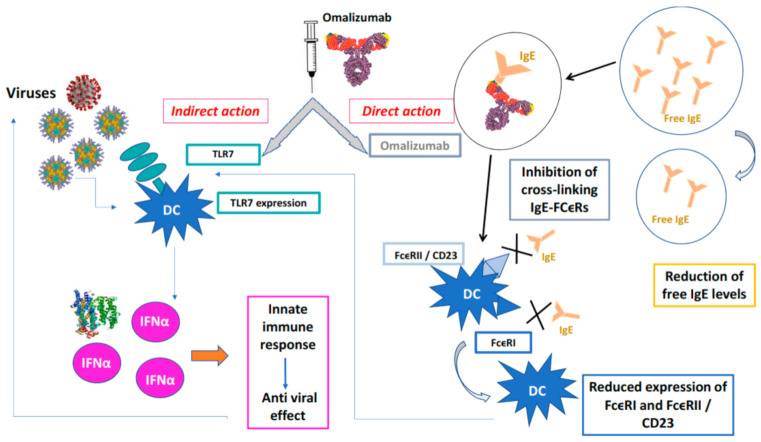
Protective action of omalizumab towards viral infections.Omalizumab acts directly forming IgE / anti-IgE immune complexes that prevent the interaction of IgE with high affinity (FcϵRI) and low affinity (FcϵRII / CD23) membrane receptors of PD cells, indirectly, PD cells binds viruses by TLCR7 that release INFα and activate innate immune response. X is the blockade of binding on the receptor.

**Table 1 biomedicines-09-00348-t001:** Recommendations for the use of biologics during COVID-19 pandemic.

Patient	Recommendation	Administration
Uninfected patients	Continued with terapy	Home-use
Infected patients	Maintained or temporarily suspended terapy	Home-use

**Table 2 biomedicines-09-00348-t002:** Studies conducted on patients with moderate-to-severe allergic asthma.

Author	Study Population	Study Design	Observations/Results
Teach et al. [14] *J. Allergy Clin. Immunol*. **2015**	453 asthmatic children aged 6 to 17 years with 1 or more recent exacerbations.	A 3-arm, randomized, double-blind, double placebo-controlled, multicenter clinical trial.	Adding omalizumab before return to school reduces fall asthma exacerbations, particularly among those with a recent exacerbation.
Novak, N et al. [6] *Immunology.* **2020**	161 patients aged 6–17 years with allergic asthma.	A prospective, observational cohort study.	Treatment with omalizumab resulted in a greater reduction in the severity of HRV-induced exacerbations than in those treated with ICS.
Trial on adult and adolescent patients:			
Wark, PAB et al. [52] *Am. J Respir. Crit. Care Med*. **2018**	10 adult patients with severe allergic asthma not controlled by maximally dosed ICS/LABA therapy.	A prospective, observational cohort study.	Omalizumab treatment improves the innate antiviral response to influenza A and HRV by increasing the production of IFN-α and IFN-λ.

## Data Availability

Not applicable.

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
