# Peer review of "Immunological Aspects Related to Viral Infections in Severe Asthma and the Role of Omalizumab"

_biomedicines, 2021, doi:10.3390/biomedicines9040348_

Round 1

Reviewer 1 Report

An interesting review on the role of the immunological aspect of viral infections and asthma and the role of omalizumab. Although the argument is broad, I enjoyed the article; I have some queries.

A materials and methods section, explicating what databases you used to perform this review and what keywords were used would in my opinion be useful.

Reference 37 is the same as 43...please check and correct.

Page 2 in the chapter "Immune response to respiratory viruses and sars cov" after reference 16, you should add: "Eosinophilic cells blood count seems to have a major role in COVID‐19 diagnosis and prognosis. Eosinopenia has been associated in up to 81% of cases and was proposed as a possible diagnostic marker for the disease. Persistent eosinopenia was associated with higher mortality." and cite an article such as :doi: 10.1111/dth.13681

Thank You

Author Response

To the Editor

Thank you for considering our manuscript entitled "Immunological aspects related to viral infections in severe asthma and the role of omalizumab".

According to the comments and suggestions of the Editor and Reviewers, we have extensively revised the manuscript and responded point-by-point to the comments, as listed below. We are submitting two versions of the manuscript: one clean revised version and a tracked version with visible amendments.

We resubmit this modified manuscript to Biomedicines, hoping it is now acceptable for publication in the journal.

Yours Sincerely,

The corresponding author, on behalf of co-authors

Dr. Francesco Menzella

Reviewer 1

Comments and Suggestions for Authors

An interesting review on the role of the immunological aspect of viral infections and asthma and the role of omalizumab. Although the argument is broad, I enjoyed the article; I have some queries.

Response: Answer: We greatly appreciate the reviewer's comment.

A materials and methods section, explicating what databases you used to perform this review and what keywords were used would in my opinion be useful.

Response: We have included a "Methods" section explaining which databases we used to perform this review and which keywords were used, as requested.

Reference 37 is the same as 43...please check and correct.

Response: Thanks to the reviewer for the query. We have replaced the duplicate reference 43.

Page 2 in the chapter "Immune response to respiratory viruses and sars cov" after reference 16, you should add: "Eosinophilic cells blood count seems to have a major role in COVID‐19 diagnosis and prognosis. Eosinopenia has been associated in up to 81% of cases and was proposed as a possible diagnostic marker for the disease. Persistent eosinopenia was associated with higher mortality." and cite an article such as :doi: 10.1111/dth.13681

Thank You

Response: We appreciate the reviewer's suggestion. We have added the suggested phrase and reference.

Reviewer 2

Comments and Suggestions for Authors

  1. Very interesting and actual topic

Response: We greatly appreciate the reviewer's comment.

  1. Two small corrections : 
  • Introduction – Respiratory viruses ... , respiratory syncytial viruses (RSV) ..

Response: We added "syncytial" as required.

  • Page 2 , third paragraph - ..” increased inflammation respiratory tract eosinohils „... should be rephrased

Response: We rephrased the sentence.

  1. Suggestion to the authors to make more clear the apparent disjunction between great susceptibility to respiratory viral infections in asthmatic patients and protective Th2 inflammatory environment  againts viral infection.

Response: We made the concept expressed in this sentence clearer, as suggested.

  1. Particular immune mechanisms of SARS CoV2  comparing to other respiratory viruses may be useful to mention, referring mainly to the cytokine storm

Response:  We have included some hints on the immune response to SARS-CoV-2 and on the cytokine storm in severe patients, as requested.

  1. The roles of eosinophils in viral infections and mostly in COVID-19 need more data and comments.

Response: We have included some data on the role of eosinophils in infections, particularly COVID-19.

  1. Evaluation of eosinopenia in patients with severe COVID-19 should consider the groups treated with and without corticosteroids separately, due to the prompt decrease of blood eosinophils after initiation of corticotherapy. Comparative evaluation of eosinophils in the blood and airways  might be recommended.

Response: We discussed in more detail the role of eosinophils in COVID-19 and the effect of corticosteroids and immunosuppressants in modifying the immune response induced by these cells.

  1. Due to potential positive role of eosinophils in viral infections, the indication for  an anti-eosinophilic  biologic therapy during SARS CoV 2 pandemy might be questionable, since eosinopenia might increase the risk for severe forms?

Response: We have specified, also on the basis of the current indications of scientific societies, that biological drugs must be discontinued in case of SARS-CoV-2 infection. We have reported literature data demonstrating that treatment with anti-eosinophil drugs does not worsen the clinical course of COVID-19.

  1. I suggest that conclusions about the beneficial effects of biologicals in severe asthma and viral infections should refer separately to omalizumab and anti-eosinophilic therapies , due to their different target and mechanisms.

Response: In the conclusions we specified that, unlike all the other biologics for asthma currently available for which there is currently no data on the possible effect of modulating the antiviral response, omalizumab is the only biologic with proven antiviral action against the viruses responsible. upper respiratory tract infections and exacerbations of asthma.

  1. Is omalizumab indicated mostly in cases with high serum IgE and/or does it show more  benefits  in these cases? 

Response: Beyond the old INNOVATE study published in 2005, in which baseline total serum IgE was the only predictor of efficacy, subsequent data did not confirm with certainty greater efficacy of omalizumab in patients with higher serum IgE levels.

  1. The need for multidisciplinary approach in indicating and managing biologicals in allergic astma, mostly allergists, pneumologists, infectious diseases specialists and pediatricians.

Response: We thank the reviewer for this comment, which we included in the discussion.

  1. Suggested references :   Leru PMBiomarkers in asthma-interpretation and utily in current asthma management. Current Respiratory Medicine Reviews, Special Issue, 2021,17, 1-0 .

Response: we included the suggested reference in the manuscript.

Reviewer 2 Report

  1. Very interesting and actual topic
  2. Two small corrections : 
  • Introduction – Respiratory viruses ... , respiratory syncytial viruses (RSV) ..
  • Page 2 , third paragraph - ..” increased inflammation respiratory tract eosinohils „... should be rephrased
  1. Suggestion to the authors to make more clear the apparent disjunction between great susceptibility to respiratory viral infections in asthmatic patients and protective Th2 inflammatory environment  againts viral infection .
  2. Particular immune mechanisms of SARS CoV2  comparing to other respiratory viruses may be useful to mention , referring mainly to the cytokine storm
  3. The roles of eosinophils in viral infections and mostly in COVID-19 need more data and comments .
  4. Evaluation of eosinopenia in patients with severe COVID-19 should consider the groups treated with and without corticosteroids separately, due to the prompt decrease of blood eosinophils after initiation of corticotherapy. Comparative evaluation of eosinophils in the blood and airways  might be recommended .
  5. Due to potential positive role of eosinophils in viral infections, the indication for  an anti-eosinophilic  biologic therapy during SARS CoV 2 pandemy might be questionable , since eosinopenia might increase the risk for severe forms ?
  6. I suggest that conclusions about the beneficial effects of biologicals in severe asthma and viral infections should refer separately to omalizumab and anti-eosinophilic therapies , due to their different target and mechanisms.
  7. Is omalizumab indicated mostly in cases with high serum IgE and/or does it show more  benefits  in these cases ? 
  8. The need for multidisciplinary approach in indicating and managing biologicals in allergic astma , mostly allergists, pneumologists, infectious diseases specialists and pediatricians .
  9. Suggested references :   Leru PM . Biomarkers in asthma-interpretation and utily in current asthma management. Current Respiratory Medicine Reviews, Special Issue, 2021,17, 1-0 .
  • Leru  PM, Deleanu DM. Romanian Allergology in the actual European Context. Review. Rom J Intern Med 2015, 53, 2, 111-117.

Author Response

Reviewer 2

Comments and Suggestions for Authors

  1. Very interesting and actual topic

Response: We greatly appreciate the reviewer's comment.

  1. Two small corrections : 
  • Introduction – Respiratory viruses ... , respiratory syncytial viruses (RSV) ..

Response: We added "syncytial" as required.

  • Page 2 , third paragraph - ..” increased inflammation respiratory tract eosinohils „... should be rephrased

Response: We rephrased the sentence.

  1. Suggestion to the authors to make more clear the apparent disjunction between great susceptibility to respiratory viral infections in asthmatic patients and protective Th2 inflammatory environment  againts viral infection.

Response: We made the concept expressed in this sentence clearer, as suggested.

  1. Particular immune mechanisms of SARS CoV2  comparing to other respiratory viruses may be useful to mention, referring mainly to the cytokine storm

Response:  We have included some hints on the immune response to SARS-CoV-2 and on the cytokine storm in severe patients, as requested.

  1. The roles of eosinophils in viral infections and mostly in COVID-19 need more data and comments.

Response: We have included some data on the role of eosinophils in infections, particularly COVID-19.

  1. Evaluation of eosinopenia in patients with severe COVID-19 should consider the groups treated with and without corticosteroids separately, due to the prompt decrease of blood eosinophils after initiation of corticotherapy. Comparative evaluation of eosinophils in the blood and airways  might be recommended.

Response: We discussed in more detail the role of eosinophils in COVID-19 and the effect of corticosteroids and immunosuppressants in modifying the immune response induced by these cells.

  1. Due to potential positive role of eosinophils in viral infections, the indication for  an anti-eosinophilic  biologic therapy during SARS CoV 2 pandemy might be questionable, since eosinopenia might increase the risk for severe forms?

Response: We have specified, also on the basis of the current indications of scientific societies, that biological drugs must be discontinued in case of SARS-CoV-2 infection. We have reported literature data demonstrating that treatment with anti-eosinophil drugs does not worsen the clinical course of COVID-19.

  1. I suggest that conclusions about the beneficial effects of biologicals in severe asthma and viral infections should refer separately to omalizumab and anti-eosinophilic therapies , due to their different target and mechanisms.

Response: In the conclusions we specified that, unlike all the other biologics for asthma currently available for which there is currently no data on the possible effect of modulating the antiviral response, omalizumab is the only biologic with proven antiviral action against the viruses responsible. upper respiratory tract infections and exacerbations of asthma.

  1. Is omalizumab indicated mostly in cases with high serum IgE and/or does it show more  benefits  in these cases? 

Response: Beyond the old INNOVATE study published in 2005, in which baseline total serum IgE was the only predictor of efficacy, subsequent data did not confirm with certainty greater efficacy of omalizumab in patients with higher serum IgE levels.

  1. The need for multidisciplinary approach in indicating and managing biologicals in allergic astma, mostly allergists, pneumologists, infectious diseases specialists and pediatricians.

Response: We thank the reviewer for this comment, which we included in the discussion.

  1. Suggested references :   Leru PMBiomarkers in asthma-interpretation and utily in current asthma management. Current Respiratory Medicine Reviews, Special Issue, 2021,17, 1-0 .

Response: we included the suggested reference in the manuscript.

Round 2

Reviewer 1 Report

The authors responded to all queries. The paper is eligible to be published.

Author Response

Thank you for comments and for accepting our manuscript. According to the comments, we have revised the manuscript to improve English as requested. We are submitting two versions of the manuscript: one clean revised version and a tracked version with visible amendments.